# Clinically Relevant β-Lactam Resistance Genes in Wastewater Treatment Plants

**DOI:** 10.3390/ijerph192113829

**Published:** 2022-10-24

**Authors:** Izabela Waśko, Aleksandra Kozińska, Ewa Kotlarska, Anna Baraniak

**Affiliations:** 1Department of Biomedical Research, National Medicines Institute, Chelmska 30/34, 00-725 Warsaw, Poland; 2Genetics and Marine Biotechnology Department, Institute of Oceanology of the Polish Academy of Sciences, Powstancow Warszawy 55, 81-712 Sopot, Poland

**Keywords:** antibiotic resistance genes, wastewater treatment plant, β-lactamase, extended-spectrum β-lactamase, carbapenemases

## Abstract

Antimicrobial resistance (AMR) is one of the largest global concerns due to its influence in multiple areas, which is consistent with One Health’s concept of close interconnections between people, animals, plants, and their shared environments. Antibiotic-resistant bacteria (ARB) and antibiotic-resistance genes (ARGs) circulate constantly in various niches, sediments, water sources, soil, and wastes of the animal and plant sectors, and is linked to human activities. Sewage of different origins gets to the wastewater treatment plants (WWTPs), where ARB and ARG removal efficiency is still insufficient, leading to their transmission to discharge points and further dissemination. Thus, WWTPs are believed to be reservoirs of ARGs and the source of spreading AMR. According to a World Health Organization report, the most critical pathogens for public health include Gram-negative bacteria resistant to third-generation cephalosporins and carbapenems (last-choice drugs), which represent β-lactams, the most widely used antibiotics. Therefore, this paper aimed to present the available research data for ARGs in WWTPs that confer resistance to β-lactam antibiotics, with a particular emphasis on clinically important life-threatening mechanisms of resistance, including extended-spectrum β-lactamases (ESBLs) and carbapenemases (KPC, NDM).

## 1. Introduction

Antibiotics are widely used to prevent and treat infections in humans, animals, and plants, but their high and incorrect consumption have made them increasingly ineffective due to antimicrobial-resistant microorganisms emerging and spreading globally. Thus, antimicrobial resistance (AMR) was announced by the World Health Organization (WHO) as one of the top global public health threats facing humanity [1]. Some Gram-negative bacteria, such as carbapenem-resistant *Pseudomonas aeruginosa*, *Acinetobacter baumannii,* and *Enterobacterales* resistant to third-generation cephalosporins and carbapenems are considered to be of particular importance, and the WHO and Centers for Disease Control and Prevention (CDC) included them in the group of critical pathogens due to the fact that they are a major cause of nosocomial infections with high morbidity and mortality [2,3]. Systematic analysis estimated 4.95 million deaths associated with bacterial AMR in 2019 and indicated β-lactam-resistant (mainly to third generation cephalosporins and carbapenem) bacteria as the major cause of death [4]. Tremendously dangerous microorganisms accumulate various AMR mechanisms that leads to their multi-drug resistance (MDR), extensive-drug resistance (XDR), or even pan-drug resistance (PDR), leaving few, one, or no therapeutic options left, respectively. Consequently, infections caused by such bacteria carry an extremely high risk of death [5,6].

AMR is ubiquitous, associated with agriculture and livestock, medical, and veterinary settings, but it is also observed in many aquatic environments, which is in line with One Health’s concept (available online: https://www.cdc.gov/onehealth/index.html accessed on 23 May 2022) of close interconnections between people, animals, plants, and their shared environments (Figure 1) [7].

Many aspects related to geographic location, socio-economic level, climate, antibiotic consumption, and the technology of the treatment process affect the abundance of antimicrobial resistance genes (ARGs), the bacteria carrying them (antibiotic-resistant bacteria, ARB), and their dissemination in the environment [8,9]. One of the factors contributing to the scale and speed of AMR spreading is the fact that high amounts of antibiotics get into sewage and, consequently, into wastewater treatment plants (WWTPs). Although the applied technology and treatment methods are constantly being improved and developed, they are still insufficient to eliminate antibiotics, ARB, and ARGs completely. Moreover, the presence of antibiotics in sub-inhibitory concentrations creates conditions for selective pressure, and additionally, other factors present in sewage, such as pesticides, detergents, and heavy metals, stimulate the co-selection of resistant strains [10,11,12,13,14,15,16].

The genetic background of the AMR transmission process is of great importance. Resistance mechanisms are genetically based and linked with many genes localized on a bacterial chromosome or, what is more dangerous, on mobile genetic elements (MGEs). The genes encode enzymes, proteins that are involved in many processes, for example inactivating antibiotics or modifying their structure, alterating drug target sites, modifying the outer membrane structure that inhibits antibiotic penetration into the cell, or the active removal of the chemotherapeutics from the cell. Due to their location on MGEs, they pose a big risk to be transferred between bacteria of the same or different species through conjugation, transduction, or transformation [17,18]. Bacteria interacting with each other and exchanging genes by horizontal gene transfer (HGT) may lead to situations wherein previously sensitive and nonpathogenic strains may get resistance determinants and become virulent or reservoirs of ARGs for further transmission. These microorganisms, as well as resistance genes, may be discharged from WWTP systems into natural water bodies like lakes, rivers, and seas [19,20,21,22,23,24], which plays an important role in their further dissemination into human, animal and plant populations [25,26,27]. Therefore, it is believed that the WWTPs are reservoirs of ARGs, so-called “hotspots”, and one of the sources of spreading AMR, especially clinically relevant ARGs [28,29,30,31].

A great effort has been made to fight AMR and many global strategies have been taken, including developing new drugs and vaccines, improving the diagnostics of resistance mechanisms, the rational use of antibiotics, infection prevention and control, and developing new technological methods for the treatment and disinfection of wastewaters [32,33,34,35]. The monitoring of AMR and identifying the migration routes of bacteria with important mechanisms of resistance in the environment is also crucial and fundamental [36]. It may help to obtain knowledge about actual epidemiological situations, the origin of ARGs, mechanisms of spreading AMR, and transmission routes, which are essential for taking appropriate actions to prevent this phenomenon. Such surveillance studies concern the occurrence of not only resistant bacteria in ecosystems but also the occurrence of resistance genes that are easily and efficiently transmissible [16,30,31,34,37,38,39,40].

Zhuang et al. analyzed PubMed publications from the last 30 years (1990–2020) concerning reports of ARGs in the environment and showed that, on all continents, the highest frequency was related to genes encoding β-lactamases, enzymes that inactivate β-lactams, the most-used group of antibiotics [41]. Therefore, this paper aimed to present available research data on the identification of β-lactamase genes in WWTPs.

For this manual review of articles from the last decade, studies of β-lactamase genes in wastewater samples and from bacteria isolated from these type of samples were analyzed, including direct WWTP (i.e., influent, sewage sludge, effluent) and WWTP-related samples (i.e., air near bioreactors, discharge points). All of the research described below is summarized in detail in Table 1, where information about the type of tested samples, stages of the treatment process, methodology used, and detected variants are included. The reviewed studies were linked to municipal/urban WWTPs; however, if the authors involved additional information about the type of collected wastewater, it was noted.

## 2. β-Lactams and β-Lactamases—Background

Among the many antimicrobial drugs available, the group of β-lactams is one of the most important and most widely used in the treatment of bacterial infections, not only as the first choice, but above all as the last-choice drugs (available online https://www.ecdc.europa.eu/en/antimicrobial-consumption/surveillance-and-disease-data/database, accessed on 23 May 2022) [42]. β-lactams are classified based on chemical structure and the target of action. The common characteristic is the presence of the three carbon and one nitrogen ring (β-lactam ring). Depending on the modifications, different groups are distinguished. Generally, there are penicillins (natural penicillins, aminopenicillins, carboxypenicillins, and ureidopenicillins), cephalosporins (divided into five classes called generations), carbapenems, and monobactams.

All β-lactam antibiotics have a common mechanism of action, which is inhibition of the bacterial cell walls’ synthesis. They block the activity of bacterial enzymes, transpeptidases known as penicillin-binding proteins (PBPs), involved in the last stage of peptidoglycan synthesis, thus inducing a loss of viability and the lysis of bacterial cells. The modification of the PBPs’ structure may lead to a reduced affinity for β-lactams, which is the major pathway for β-lactam resistance among Gram-positive bacteria but is not very common for Gram-negative bacteria [5,43,44]. The other mechanisms of resistance, detected mainly in Gram-negative bacteria, are related to cell membrane modulations, including: (i) the reduction or loss of outer membrane porins that restrict the entry of antibiotic into the cell or (ii) the expression/overexpression of the efflux pump that allows the effective removal of the antibiotic from the cell. Examples are AcrAB-TolC-type pumps, described in clinical isolates of *Klebsiella pneumoniae*, and MexAB-OprM pumps, reported in *P. aeruginosa* [45]. Finally, the most common mechanism in Gram-negative bacteria, and relatively rarely found in Gram-positive bacteria, is the production of β-lactamases, enzymes that hydrolyze β-lactam antibiotics making them ineffective. These enzymes are critical, causing hard-treated human infections (urinary tract infections, bloodstream infections, wound infections, and pneumonia), especially caused by *P. aeruginosa*, *A. baumannii,* and *Enterobacterales*; thus, this paper focuses on them.

Two classification systems of β-lactamases are used. The structural one, based on the amino acid sequence of the enzyme, groups β-lactamases into 4 classes, A, B, C, and D, of which A, C, and D are β-lactamases with serine in the active center, while class B uses zinc cations as cofactors of the hydrolysis reaction (metallo-β-lactamases, MBL). Another classification scheme, functional, is based on substrate hydrolysis profiles and the inhibitor profile, distinguishing four main functional groups, 1–4. Group 1 includes cephalosporinases and cephamycinases, which are very weakly inhibited or uninhibited by clavulanic acid; group 2 is very extensive and diverse, with different substrate spectra, mostly inhibited by clavulanic acid. Group 3 additionally hydrolyzes carbapenems, but their activity is inhibited by EDTA, and group 4 are penicillinases weakly inhibited by clavulanic acid. Both classification systems of β-lactamases correlate well with each other. All of the enzymes that make up functional group 1 are structural class C; group 2 contains β-lactamases of classes A and D, and group 3 corresponds to class B [46,47].

The general scenario of β-lactamase evolution was stimulated by the mass use of β-lactam antibiotics, as shown in Figure 2. It reveals a kind of “race” between pathogenic microorganisms and the pharmaceutical industry, which develops ever newer “generations” of β-lactams, as well as the adaptation of bacteria to environments in which the selection pressure of “older” and “newer” drugs accumulates. Shortly after the introduction of penicillins (benzylpenicillins) into therapy in the 1940s, the emergence and rapid growth of β-lactamase-producing strains of *Staphylococcus aureus* was observed. The first cephalosporins and broad-spectrum penicillins, used since the early 1960s, mainly against β-lactamase-producing *S. aureus* and/or Gram-negative bacilli, contributed to the emergence of new resistance mechanisms. Among other things, this resulted in the selection of *Enterobacterales* producing plasmid-encoded broad-spectrum β-lactamases. In turn, the intensive use of oxyimino-β-lactams since the early 1980s has led to the selection of new mechanisms of acquired resistance. This resistance is mainly related to the production of extended-spectrum β-lactamases (ESBLs) and acquired AmpC and includes phenomena such as the derepression or overexpression of AmpC. Finally, the bacterial response to the introduction of carbapenems has been the emergence of strains producing acquired carbapenemases such as MBLs and some class A and D enzymes.

All β-lactamases are encoded by *bla* genes and located on the bacterial chromosome or MGEs like plasmids, transposons, and integrons with gene cassettes. Bacteria can acquire ARGs by horizontal gene transfer, HGT, which enables the exchange of genetic material between commensals, environmental species, and pathogenic bacteria; therefore, HGT is considered the main method of antibiotic resistance dissemination [18].

## 3. Methods of AMR and ARGs Analysis in Environmental Samples

The monitoring and evaluation of ARB in water environments use various methods, generally divided into two groups: culture-dependent and culture-independent. The first one is based on traditional microbiological methods used in clinical surveillance, requiring strains isolated from the environmental samples (determining: taxonomy, antibiotic susceptibility profiles, resistance mechanisms). To evaluate the level and mechanism of resistance carried by bacteria, the disk diffusion method and minimum inhibitory concentration (MIC) assays are used, according to the European Committee on Antimicrobial Susceptibility Testing (EUCAST; available online: www.eucast.org) and the Clinical & Laboratory Standards Institute (CLSI; available online: www.clsi.org). Analysis of the AMR patterns of strains may provide information about multidrug resistance. Bacterial conjugation assays are also conducted to confirm the transferability of selected genes. Time-consumption is the main limitation of such methods, because they require pure bacterial cultures, which may be troublesome or even unavailable for slow-growing bacteria. Additionally, breakpoints for antibiotic susceptibility tests may be applied to a narrow spectrum of pathogens detected in wastewater, only to clinical bacteria for which recommendations are available.

Therefore culture-independent, DNA-based methods were developed and, in recent years, have become extensively used. Molecular techniques, including nucleic acid amplification (polymerase chain reaction, PCR) and DNA sequencing, are successfully used for the analysis of direct environmental samples but are also widely used for the molecular analysis of isolated strains for the detection of genetic resistance determinants (ARGs, MGEs) and/or molecular typing methods to define genetic relatedness between isolates with clinical and environmental origin (multi-locus sequence typing, MLST; phylogrouping; pulsed-field gel electrophoresis, PFGE). Some studies focus on defining the efficiency of the treatment process; therefore, quantitative PCR (qPCR) is used to determine the number of selected gene copies/mL (absolute abundance) and/or the number of copies normalized to 16S rRNA copies (relative abundance). The developing metagenomics approaches that use various techniques of molecular biology deserve special attention. Metagenomics allows us to explore the biodiversity of a population of microorganisms and the identification of the present genes, as well as detecting new ones and determining their functions and analyzing their origin and the transfer and dissemination of ARGs between species [16,30,48,49,50,51,52]. Most results of the metagenomics approaches in sewage contain the data of the resistance genes present in different stages of the treatment process; correlations with various factors, like heavy metals, MGEs, and antibiotics, on the ARGs’ occurrence and abundance; and their transfer and removal efficiency in different types of treatment processes and disinfection. The intensification of metagenomics research concerning AMR in WWTPs has been significant in recent years; however, due to the different approaches, different goals of the research, variety of tested samples, and types of WWTPs, the obtained results may be difficult to compare; thus, the procedures should be standardized. However, the analysis of the data gives an overall picture and information on general trends concerning the spread of antibiotic resistance [36,53].

## 4. Clinically Significant β-Lactam Resistance Genes in Wastewater Treatment Plants—The Occurrence and Distribution

According to the β-lactamase database (available online: BLDB; http://bldb.eu/ accessed on 23 May 2022), these enzymes constitute a very heterogeneous group with more than 7000 genetic variants identified. Within each of the four classes (A, B, C, and D), β-lactamases of particular clinical importance can be distinguished. These are detected consistently in environmental niches, including WWTPs (Figure 3) [41].

### 4.1. Class A β-Lactamases

Class A β-lactamases are serine proteases that hydrolyze, on various levels, penicillins, monobactams, cephalosporins, and carbapenems and may be inhibited by β-lactamase inhibitors (e.g., clavulanic acid, sulbactam, tazobactam). It is the most diverse group, consisting of the enzymes with various spectra of hydrolysis, generally divided into: (i) a group with a narrow spectrum, e.g., carbenicillin-hydrolyzing β-lactamase (CARB) and *Pseudomonas aeruginosa* β-lactamase (PSE); (ii) a group with extended spectrum (ESBL) enzymes that originated from the first group but modified due to point mutations within the genes encoding them, which results in broadening their spectrum of hydrolyzing, e.g., cefotaximase-München-lactamase (CTX-M), Temoniera-lactamase (TEM), and sulfhydryl variable-lactamase (SHV); and (iii) a group with extremely extended spectrum including carbapenems—antibiotics of the last resort, e.g., Guiana extended-spectrum (GES), Klebsiella pneumoniae carbapenemase (KPC), Serratia marcescens enzyme (SME), and Serratia fonticola carbapenemase A (SFC-1) [54]. Among all, ESBL- and carbapenemase KPC-producing bacteria attracted the largest amount of clinical concern. Both TEM- and SHV-type ESBLs were described throughout the United States (US) and Europe in the late 1980s and 1990s, with specific variants noted to be regional in distribution [55,56]. The prevalence of these enzymes has now diminished at the same time as the worldwide dissemination of isolates producing CTX-M-type β-lactamases [57,58]. Once limited to hospital settings, ESBL-producing isolates quickly expanded into nursing homes and community settings as well [59,60]. The propagation of *Enterobacterales*-possessing ESBLs has had a significant impact on the choice of empirical antimicrobial therapy, driving the use of carbapenems in many institutions and resulting in increased resistance to carbapenems [61]. KPC carbapenemase has been extensively reported in *K. pneumoniae*, and it is endemic in the US but also in Latin America, China, Israel, and some European countries, such as Greece and Italy [62,63,64].

#### 4.1.1. Class A β-Lactamases—Occurrence and Variability in WWTPs-Linked Samples

Due to the global spread of class A β-lactamases, it is a commonly, or even predominantly, detected group in WWTPs (Table 1). In a multi-national study of WWTPs from Denmark, Spain, and the United Kingdom (UK) with high-throughput qPCR used, these β-lactamases was leading, accounting for 70% of all detected *bla* genes [65]. Among them, the most relevant were two groups linked with ESBL and KPC enzymes. It is noteworthy that, among the ESBL group, the most common in clinical settings and in various wastewater sources is CTX-M encoded by *bla*_CTX-M_, carried mainly by *Enterobacterales* [66]. In this review, *bla*_CTX-M_ was detected in the majority of included studies and, in many, had the highest prevalence [67,68,69,70,71,72,73,74,75,76,77,78,79,80,81]. However, *bla*_SHV_ and/or *bla*_TEM_ were found frequently as well [15,82,83,84,85]. In some studies, *bla*_TEM_ was predominant, e.g., in an Irish study [86], as well as in Colombia [87], Poland and Portugal [9,88], Belgium [89], the US [90], and Africa [91]. Another significant group representing the KPC family encoded by *bla*_KPC_ genes was detected in numerous WWTPs from European [65,69,89,92,93,94,95,96,97,98,99,100,101], as well as from American [90,102,103,104], African [72,91,105], and Asian countries [106,107]. Moreover, analysis of reviewed articles, especially those using developed techniques as high-throughput qPCR, whole-genome sequencing, or metagenomics, shows a high variety of detected genes of the discussed β-lactamases, not only representing *bla*_CTX-M_, *bla*_SHV_, *bla*_TEM_, and *bla*_KPC_ families, but also others less frequently associated with public health, i.e., BEL, cfxA, GES, PER, SME, VEB, and others [65,92,96,102,107,108,109,110,111].

Environmental studies based on the analysis of bacterial strains during the treatment process most often concern the most critical pathogens posing the greatest threat, mainly *Enterobacterales*. In the reviewed literature, the predominantly tested and detected species among this bacteria family were *Escherichia coli* and *K. pneumoniae* [67,68,71,72,74,77,78,79,81,83,85,99,112,113,114,115,116,117,118,119,120,121,122,123,124,125]; however, different species of *Citrobacter* spp., *Enterobacter* spp., *Pseudomonas* spp., *Aeromonas* spp., or others were noted as well [71,76,80,87,92,102,104,109,110,121,126,127,128].

It is noteworthy that antibiotic susceptibility testing of the studied ESBL-producing strains isolated from the WWTPs confirms a high percentage of multi-drug resistance. It was also noted that these bacteria may survive the treatment process and that the WWTPs were unable to eradicate them completely. Generally, the number of MDR isolates decreased during the treatment, but for some, their proportion was still significant in effluents, in some even higher than in influent samples [70,71,73,86,88,92,94,97,99,118,119,123,125,129,130]. Moreover, analyzing downstream river or marine samples where final effluents are released, MDR isolates carrying ESBL enzymes were commonly detected [20,79,88,92,130].

Molecular typing concerning bacteria isolated from WWTPs confirmed high genetic relatedness between bacteria from WWTPs and human- and animal-associated sources, as well as the presence of clinically important lineages such as pandemic ST131 *E. coli* in WWTPs-related samples. Liedhegner et al. compared *E. coli* isolated from samples of various environmental compartments from one geographic area (clinical samples, hospital wastewater, and WWTP). The data including antibiotic resistance, virulence, and ESBL gene profiles confirmed high phenotypic and genotypic similarity across strains of these different origins and demonstrated potential health risks related to ESBL transmission [125]. An interesting study conducted by Raven et al. showed genetic relatedness between *E. coli* isolated from 20 WWTPs in the UK, livestock farms, retail meat, and isolates responsible for human blood infections. The genomic analysis of i.e., ESBL-producing isolates revealed that the three most common sequence types (STs) associated with bloodstream infections (ST131, ST73, and ST95) and the specific and most common for livestock (ST10) were found in wastewater samples [120]. In many other studies, human-associated, multidrug-resistant, and highly virulent clone ST131 *E. coli* was detected in WWTP samples as well [75,87,113,131,132,133,134].

#### 4.1.2. Class A β-Lactamases—Removal during the Treatment Process

Concerning the removal of class A β-lactam ARGs, there is no universal target panel in qPCR studies; however, it has been noted that, although the WWTPs could effectively eliminate examined genes, their abundance was still reported in effluents and receiving water bodies. For example, in the study of Schages et al., strains harboring *bla*_CTX-M_ were isolated from the effluent [123], as well as in a Japanese study wherein strains possessing ARGs belonging to the *bla*_CTX-M-1_, *bla*_CTX-M-9_, *bla*_TEM_, and *bla*_SHV_ families survived even after sterilization [124]. Other studies reported similar results of the ARGs’ presence in effluent samples [108,135,136,137,138]. In Polish research from Kozieglowy, it was noticed that the wastewater treatment process leads to a significant increase in the relative abundance of *bla*_TEM_ and *bla*_GES_ genes, while the abundance of *bla*_KPC_ decreased. Finally, the removal efficiency of ARGs was the least for *bla*_GES_ (94.8%) and *bla*_CTX-M_ (95.3%), while for other genes, it was >98% [69]. In another study, the presence of *bla*_KPC_ was completely eliminated even after the first mechanical procedure [93]. In a Chinese survey comparing bacteria carrying *bla*_CTX-M_, *bla*_SHV_, and *bla*_TEM_, isolated from influent and effluent, higher prevalence was noted in influent samples, except for *bla*_CTX-M_, which was more frequently detected in effluent samples [129]. Significant differences between influent and effluent were described in a Romanian investigation and concerned *bla*_SHV-100, -145_, which were decreased during treatment [85]. Interestingly, Neudorf et al. analyzed 3 WWTPs in Arctic Canada and noted a decrease of *bla*_TEM_ abundance in two sites with a passive system and no significant changes for a third WWTP with a mechanical system. Moreover, no differences were found for *bla*_CTX-M_ in all treatment plants [139]. A Spanish study by Rodriguez-Mozaz et al. demonstrated an increased frequency of *bla*_TEM_ during the treatment process [140], while in a study of three WWTPs from Finland and Estonia, no significant changes were noted for *bla*_CTX-M-32_, unlike *bla*_SHV-34_, of which the relative concentration was increased in effluent samples but only in one tested WWTP [141]. Comparable data with similar *bla*_CTX-M_ and *bla*_TEM_ concentrations in influent and effluent samples were obtained in a study of five WWTPs in Tunisia; however, the abundance of the genes was higher in the effluent in a WWTP receiving additional hospital wastewater [142]. The occurrence of class A β-lactamases ARGs was also detected in downstream river samples whence final effluents were discharged, e.g., in a multi-national study including sixteen WWTPs from ten European countries [101], in a study conducted by Zieliński et al., wherein the predominant *bla*_TEM_ was noted in receiving river water samples [15], and in a study performed by Osińska et al., wherein the presence of *bla*_SHV_ and *bla*_TEM_ in receiving river samples was confirmed [84].

WWTPs pose a health risk, not only because treated wastewater containing AMR genes or MDR bacteria are transferred into surface water bodies, but also because these pollutants are discharged into the air surrounding WWTPs through bioaerosol generated from bioreactors [15,68]. The study of the carriage of ESBL-producing *Enterobacterales* in WWTP workers and surrounding residents shows that these groups are much more like to acquire bacteria harboring the ESBL mechanism [25], thus confirming the direct influence of WWTPs on spreading ARGs into air. The contribution of WWTPs’ bioaerosols in ARGs and ARB propagation into air and different environments is commonly investigated [143,144,145,146]. For example, Gaviria-Figueroa et al. studied bioaerosol samples collected downwind from sludge aeration tanks and showed a significant presence of clinically relevant class A β-lactamases, along with other classes of these enzymes and different antibiotic groups [147].

### 4.2. Class B β-Lactamases

Class B β-lactamases consist of a wide variety of metallo-β-lactamases (MBLs), enzymes able to hydrolyze almost all β-lactams: penicillins, cephalosporins, clinically available β-lactamase inhibitors, and carbapenems, except monobactams. They use zinc ions for activity, hence the name “metallo-” and susceptibility to metallic ion chelators like EDTA. Numerous variants are distinguished and grouped into three subclasses, among which the most widespread MBLs are imipenem-resistant *Pseudomonas* (IMP), Verona integron-encoded metallo-β-lactamase (VIM), and New Delhi metallo-β-lactamase (NDM), all representing subclass B1 [54,148,149,150]. MBLs initially detected in *P. aeruginosa* are frequently found nowadays in *K. pneumoniae* and other *Enterobacterales* [62,63,64]. IMP carbapenemases mainly contribute to carbapenem resistance in Japan, as well as in other regions of Southeast Asia and Australia [151,152,153]. Although they have not spread extensively throughout the rest of the world, they are being reported more frequently in Middle-Eastern countries [154]. VIM MBLs are identified more frequently than IMP enzymes [155]. Initially, they spread rapidly throughout southern Europe with major outbreaks of VIM-producing *P. aeruginosa* reported in Italy and Greece in 2006, followed by outbreaks of VIM-producing *K. pneumoniae* [156,157]. Today, they are found globally, mainly in *K. pneumoniae* and *E. cloacae* complex strains [151]. Among the major types of MBLs, the NDM-type variants are especially associated with *Enterobacterales*. The first NDM was identified in 2008 in a *K. pneumoniae* isolate from a patient in Sweden who had arrived from India [158]. The Indian subcontinent, the Balkans, and the Middle-East/North Africa are considered to be the main NDM reservoirs [62,63]. An extremely wide spectrum of metallo-β-lactamases and the fact that isolates possessing MBL genes often simultaneously harbor other antibiotic resistance genes make these organisms an urgent public health threat. Although there is substantial geographic variability in the prevalence of MBL enzymes, they are noted worldwide and the speed of their dissemination is alarming, especially NDM enzymes [44,54,159,160,161].

#### 4.2.1. IMP and VIM β-Lactamases in WWTPs-Linked Samples

As with the previously discussed ARGs, the environment plays a role in the transmission of *bla*_IMP_ and *bla*_VIM_ encoding MBLs enzymes with clinical importance, IMP and VIM, respectively (Table 1). Although the majority of reports focus on hospital wastewater, these genes were detected also in samples of wastewater treatment plants from the US [82,102,103,147], Canada [104], China [70,82], and Singapore [107] as well as from many European countries, such as Sweden [96,109], Switzerland [99], the UK [128], Germany [100,123,136,162], Poland [69,92,93,163], Slovakia [115], and Romania [94]. A multi-national study concerning urban WWTPs in Denmark, Spain, and the UK showed the permanent presence of *bla*_VIM_ during the treatment process even in downstream river samples, in contrast to other tested genes, which were reduced under a detectable level [65]. Interesting results were presented by Khan et al., who compared Klebsiella oxytoca strains isolated from clinical sources (hospital wastewater) and the river receiving effluents from WWTP in Örebro, Sweden. Results obtained for two selected strains—the same antibiotic susceptibility patterns, antibiotic resistance gene profiles (i.e., *bla*_VIM-1_, *bla*_OXA-10_, *bla*_ACC-1_), MLST type, furthermore phylogenetic relationship based on core genome single nucleotide polymorphism (SNP) analysis, and core genome MLST—suggest the transfer of K. oxytoca-producing carbapenemases from the hospital setting to the aquatic environment, which may pose a threat to the community [164].

#### 4.2.2. New Delhi Metallo-β-Lactamase (NDM) in WWTPs-Linked Samples

According to epidemiological data, NDMs seem to pose the greatest threat among class B β-lactamases. Genes encoding them were noted in many aquatic environments, including animal production wastewaters, industrial, domestic sewage, tap water, surface water, and groundwater. However, hospital wastewater is considered to be a major source of *bla*_NDM_ variants [165,166,167]. As the geographical origin of NDM-producing bacteria is India, multiple publications detecting *bla*_NDM_, especially in hospital sewage, come from India [168,169,170], together with other Asian [108,171,172,173] and African countries [105,174]. Interesting results were reported by Marathe et al., who studied hospital wastewater from Mumbai, India. Shotgun metagenomics revealed the presence of β-lactamase genes encoding clinically important MBLs, such as NDM, VIM, and IMP with *bla*_NDM_ as the most common carbapenemase-encoding gene. Additionally, 27 unique MBL genes not known yet were detected, which showed the huge potential of the metagenomic approach [175]. However, NDM-lactamases in Asian countries were not only detected in hospital sewage samples (Table 1). Analysis of rivers and sewage treatment plants in five Indian states also showed an abundance of *bla*_NDM_ [77]. Similarly obtained data from southwest China showed a wide distribution of *bla*_NDM_ in hospital sewage, WWTP effluent, and river samples. Interestingly, the gene was found in many different bacterial species belonging to *Enterobacterales*, genus Acinetobacter, and *Pseudomonas* [176]. The data from northern China [177,178] and Saudi Arabia [179] also confirm the presence of *bla*_NDM_ in WWTP samples. *bla*_NDM_ has spread globally, and several variants were noted not only in India and China but in many other countries in various water samples, including those from WWTPs and the surface waters of WWTP discharge points in the UK [128], Belgium [89], Switzerland [99], Germany [100], Poland [69,163], the Czech Republic [180], Romania [85,94], Spain [98], Africa [91,105], and the US [90,102,103]. Interesting results concern the Irish study conducted by Mahon et al. They examined the genetic relationship between NDM-possessing *E. coli* and *K. pneumoniae* (separately) cultivated from three locally linked sources: sewage samples from the collection system, freshwater streams, and clinical isolates. *E. coli* were considered indistinguishable, and *K. pneumoniae* were very closely related. These results confirm that water sewage plays an important role in the resistance transfer process [181]. Another analysis by Walsh et al. concerning public tap water and seepage water from sites around New Delhi also indicates that the environment has an undeniable influence on the propagation of NDM resistance [182].

Data regarding the wastewater treatment process show a different level of the transmission of bacteria with the NDM mechanism during the treatment process and the effectiveness of *bla*_NDM_ reduction. In a Polish urban WWTP from Kozieglowy, Makowska et al. studied β-lactamase genes in the genomes of ESBL-producing and carbapenem-resistant coliforms isolated from each stage of the treatment process. They found that *bla*_NDM_ and *bla*_VIM_ were present in all stages and that the highest frequency was recorded in isolates from effluent compared to raw sewage, which indicates that the treatment process in the mechanical–biological treatment plant is insufficient in eliminating *bla*_NDM_ and the organisms carrying them [69]. Similarly, data from two WWTPs in north China show the persistent and prevailing presence of *bla*_NDM_ even after disinfection [177] and the propagation of *bla*_NDM_ from a WWTP into its receiving river [178]. Other studies measuring absolute (copies/mL) and relative (copies/16S) abundance of *bla*_NDM_ in influent and effluent also confirm deficient reduction [98,183]. However, Divyashree et al., who studied treated and untreated effluents from hospital samples in Mangalore, South India, showed the absence of *bla*_NDM_ in treated effluents [184]. A Polish study also showed a complete reduction of *bla*_NDM_ in the treatment process, even after the initial treatment stage [93], similar to a multi-center study from Denmark, Spain, and the UK [65].

### 4.3. Class C β-Lactamases

β-lactamases belonging to class C (AmpC) confer resistance to broad-spectrum β-lactams including penicillins, monobactams, and, most of all, cephalosporins (except fourth and fifth generations). Three mechanisms of resistance are noted: (i) chromosomal resistance induced by β–lactams; (ii) derepression due to mutations in AmpC regulatory genes, which results in overexpression and the production of the enzyme at a very high level; and (iii) the presence of plasmid-mediated AmpC genes (pAmpC) that are easily transmissible, even between different species, thus posing the highest health risk among class C β-lactamases. The first pAmpC variant was identified in 1989 from *K. pneumoniae* isolated in South Korea [185]. Several families of plasmid-encoded AmpC variants were reported within the next decade, i.e., ACC, CIT (variants CMY, LAT, BIL), DHA, EBC (variants ACT, MIR), FOX, and MOX, differing in bacterial species of origin. The most commonly found among the strains responsible for human infections are ACC, CMY, and DHA enzymes encoded by *bla*_ACC_, *bla*_CMY_, and *bla*_DHA_ genes, respectively. Clinically relevant bacteria producing pAmpC enzymes are mainly *Citrobacter* spp., *Salmonella* spp., and *Shigella* spp., but they were also found in other *Enterobacterales*, including *K. pneumoniae*, *Enterobacter*
*aerogenes*, *Proteus mirabilis*, *Morganella morganii*, and *K. oxytoca* [44,47,186,187].

#### Class C β-Lactamases in WWTPs-Linked Samples

Similar to the clinical surveillance of pAmpC, environmental studies concerning wastewaters and WWTPs report the predominance of genes encoding CMY and DHA enzymes (Table 1). Kwak et al. conducted an antimicrobial resistance analysis of *E. coli* in urban and hospital wastewaters. They noticed that, among β-lactam-resistant ARB, almost all (97%) were confirmed to possess ESBL or pAmpC, and among pAmpC, all were detected as carrying the *bla*_CMY-2_ variant [116]. This variant, as well as others representing the CMY and DHA families, were detected in many other European studies of WWTPs from Germany [123,135,136], Romania [85], Sweden [96,109], Portugal [88,110], Poland [92,93], Slovakia [115], and Spain [188], as well as in studies conducted in Africa [127], North America [80,90,102,147,189], South America [87], and Asia [77,107]. Interestingly, Yim et al. investigated samples for plasmid-mediated quinolone resistance genes from a WWTP in Canada and detected the presence of qnrB4-AmpC (*bla*_DHA-1_) genes in plasmids among *Citrobacter freundii* isolates. These were almost identical to those found in pathogenic *Klebsiella* isolates. Results of SNP analysis may suggest their dissemination from WWTP strains into clinical strains, which supports that WWTPs are a source of AMR spread [189].

In the reviewed studies, AmpC genes were detected in different stages of the treatment process, as well as in surface waters related to WWTPs. Alexander et al. conducted research on 20 critical points in aquatic systems, including WWTPs, and showed that, although the abundance at individual points and sampling periods over 2 years was variable, the presence of the AmpC genes was found in all sampling sites [162]. In another study, Su et al. analyzed the AmpC genes in *Escherichia coli* from two municipal WWTPs in China and noted that AmpC was detected in all treatment stages [190]. In s multi-national study, Yang et al. used shotgun metagenomics on activated sludge samples of 15 WWTPs from China, Singapore, the US, and Canada and detected the highest abundance of AmpC genes among all tested β-lactam resistance genes. They also found very high genetic diversity of AmpC genes [82]. Generally, metagenomic studies or studies using high throughput PCR are very useful in detecting multiple variants of genes encoding AmpC and representing different families, including, i.e., FOX, MOX, MIR, ACT, and ACC [65,93,96,102,104,107,109,123,147].

Although *bla*_CMY_ and *bla*_DHA_ are the most often detected and prevalent pAmpC genes, in some studies, other variants are predominant. For example, Amador et al. showed that, among the AmpC-producing *Enterobacterales* isolated from Portuguese WWTP samples, the dominant was *bla*_EBC_, followed by *bla*_FOX_ and *bla*_CIT_ [88]; Piotrowska et al. analyzed *Aeromonas* spp. strains isolated from urban WWTPs in Warsaw, Poland, and found *bla*_FOX_ to be the most abundant, followed by *bla*_MOX_ and *bla*_ACC_ [97]. For comparison, Fadare and Okoh studied *Enterobacterales* isolated from the effluents of two WWTPs in South Africa and reported that the most predominant were *bla*_CIT_ and *bla*_ACC_, whereas *bla*_FOX_ was detected in only one isolate [72].

Due to the lower frequency and speed of spread compared to other β-lactam resistance mechanisms, AmpC enzymes do not represent such a high risk. However, they are present in WWTP samples including effluents, and as a result of plasmid-localized and HGT present during the treatment process, this group may still pose a health risk and needs to be monitored.

### 4.4. Class D β-Lactamases

According to the BLDB, class D β-lactamases, known as oxacillinases, include more than 1,000 enzymes divided into 19 groups, among which the OXA group is the most numerous and clinically relevant. Among these, carbapenem-hydrolyzing class D enzymes (CHDLs) pose the greatest risk [47]. The substrate spectrum of the variants and level of hydrolyzing may significantly differ; however, all class D β-lactamases are not inhibited by β-lactam inhibitors, and they confer resistance to the amino-, carboxy-, and ureidopenicillins [191]. Although not classical ESBLs, as defined by inhibition by clavulanate, several of the OXA-type β-lactamase variants, such as OXA-11 and OXA-14 to OXA-20, are associated with an ESBL phenotype in that they confer resistance to some of the late-generation cephalosporins [192]. Within the OXA family, only a small fraction has a functional role as a carbapenemase. Among these are OXA-23, OXA-40, and the increasingly prevalent OXA-48, with its related variants, OXA-162, OXA-181, and OXA-232 [193]. The major enterobacterial class D carbapenemase, OXA-48, was first reported in a Turkish *K. pneumoniae* isolate in 2001 [194]. Thereafter, OXA-48 and related variants have been found in almost all *Enterobacterales*, mainly in *K. pneumoniae* and *E. coli*, that spread globally, causing endemic states in the Middle East, North Africa, India, and some European countries [62,63,64].

#### 4.4.1. OXA Family β-Lactamases Carried in ARB

The reviewed approaches concerning class D β-lactamases are focused on bacterial strains carrying *bla*_OXA_ isolated from WWTP samples (Table 1). The majority of these studies confirm a *bla*_OXA_ presence in isolates from both untreated and treated samples, and the prevalent variants are *bla*_OXA-1_ and *bla*_OXA-48_. Multiple examples come from European countries: a Czech study reported ESBL-producing *Enterobacterales* carrying *bla*_OXA-1_ and isolated from effluent; globally spread MDR clones of *E. coli* ST131 and *K. pneumoniae* ST321 and ST323 harboring large FIIK plasmids with multiple antibiotic-resistance genes were found among tested strains [113]; a Spanish study detected *bla*_OXA-1_ in strains isolated from effluents of two out of 21 tested WWTPs [76]; two German studies reported the presence of *bla*_OXA-51_ and *bla*_OXA-48_ in carbapenemase-producing bacteria [100] and *bla*_OXA-58_, *bla*_OXA-48_ and *bla*_OXA-23_ in bacterial strains isolated from influent, activated sludge and effluent [123]; four Polish studies identified *bla*_OXA_ genes among ceftazidime- or meropenem-resistant bacterial strains [92], *Aeromonas* spp. strains isolated from raw sewage, activated sludge, and effluent [97], ESBL-producing *Enterobacterales* [68] and *Acinetobacter* spp. isolates [163]; an Austrian study of carbapenemase-producing *Enterobacterales* from activated sludge confirmed harboring *bla*_OXA-48_ [95]; and a study concerning the WWTP in Basel, Switzerland, where carbapenemase-resistant *Enterobacterales* and other Gram-negative bacteria isolated from municipal and hospital wastewater and WWTP receiving this sewage were compared, and identical isolates from the WWTP and wastewater samples were detected, including OXA-48-producing *E. coli* ST38 and *Citrobacter* spp. [99]. Similarly, a molecular epidemiology approach was conducted in a Romanian study. Surleac et al. detected variants of *bla*_OXA_ in *K. pneumoniae* isolated from samples of three WWTPs [85], while Teban-Man et al. compared carbapenemase-producing *K. pneumoniae* isolated from the influent and effluent of two WWTPs with and without hospital input and found that *bla*_OXA-48_ was carried by strains isolated from raw and treated samples of WWTPs collecting hospital wastewater. In the second WWTP, the gene was observed only in strains from influent. Moreover, isolates harboring *bla*_OXA-48_ were genetically typed, which showed they belonged to sequence types of high-risk clones (ST258, ST101, ST147, ST2502). These clones were associated with clinical settings and reported to be multi-drug resistant [94]. In a study of a Swedish WWTP, Gram-negative bacteria harboring *bla*_OXA_ were noted in influent, effluent, and recipient waters of the river and lake [109]. However, in a Portuguese study conducted by Araujo et al., *bla*_OXA_ was detected only in strains isolated from raw sewage samples [110]. Another Portuguese investigation of ampicillin-resistant *Enterobacterales* isolated from influent and effluent showed different results; *bla*_OXA_ was the most prevalent gene among tested ESBL-producing strains [88]. There are significantly fewer studies detecting *bla*_OXA_ in the African region and they cover Algeria, where *bla*_OXA-1_ was detected [74]; Durban, South Africa, where cefotaxime-resistant *E. coli* were studied and *bla*_OXA-1_ was found as well [91]; Eastern Cape Province, South Africa, where *bla*_OXA-1-like_ and *bla*_OXA-48-like_ variants harbored by *Enterobacterales* isolated from effluents of WWTPs were noted [72]; and Tunisia, where *C. freundii* isolate carrying *bla*_OXA-204_ [121] and *Enterobacterales* strains possessing *bla*_OXA-1_ [127] were detected. American studies concerning WWTPs also confirm *bla*_OXA_ presence in bacteria isolated from WWTP samples [80,90,102,104,119,125,133].

#### 4.4.2. OXA Family β-Lactamases in Direct WWTP Samples—Occurrence and Removal

Multiple studies report the presence of *bla*_OXA_ in direct WWTP samples and determine the concentration and relative abundance of selected gene variants to define the efficiency of the treatment process (Table 1). Comparable to previously discussed β-lactamases, bacteria producing OXA enzymes, as well as *bla*_OXA_, can be detected after the treatment process. For example, the study of two WWTPs in the Brussels region determined the relative abundance of *bla*_OXA-48_ in different stages of the treatment, as well as in samples of the river as the discharge point for the WWTP effluents. In that study, Proia et al. showed a significant increase of *bla*_OXA-48_ from influent to effluent and from upstream to downstream river samples [89]. Similarly, in Kozieglowy, a Polish WWTP, it was reported that the wastewater treatment process leads to a significant increase in the relative abundance of *bla*_OXA-48_ genes in the effluent [69], whereas in research from the Baltic Sea area, the relative abundance of *bla*_OXA-58_ was decreased in the effluent; however it was weakly significant and found only in one of the three studied WWTPs [141]. In the German study, the absolute abundance of selected *bla*_OXA_ genes was determined, and when comparing raw and treated samples from WWTP, a significant decrease was reported regarding *bla*_OXA-58_ and *bla*_OXA-48_ but not *bla*_OXA-23_ [123]. Similar results were obtained in a multi-national study of WWTPs from ten European countries, where qPCR and absolute abundance were performed for selected *bla*_OXA_ genes. It was noticeable that, among all tested β-lactamase genes, *bla*_OXA-58_ was found in all tested samples, had the highest absolute abundance, and was significantly reduced during treatment [101]. In three Swedish municipal sludge treatment plants, a metagenomics approach was conducted, and many variants of *bla*_OXA_ were detected at all stages of the treatment process. Some of them, like *bla*_OXA-48_, were consistently enriched in treated sludge compared to primary sludge [96]. Other metagenomic approaches or using qPCR provide similar results—the presence of multiple *bla*_OXA_ gene variants, including effluent samples [93,109,136,147], while others detected only single or a few variants [65,69,89,101,107,108,111,141,163]. Interestingly, in a Polish study, where *bla*_OXA_ was detected as one of the prevalent tested genes in influent and effluent samples, comparative metagenomic analysis of DNA from WWTP samples and employees’ swabs revealed the presence of similar ARGs in both types of samples with significantly higher concentrations than in control samples [15]. Other studies that report the presence of *bla*_OXA_ genes at different stages of the treatment process include the research of Yang et al., wherein activated sewage sludge from 15 WWTPs was tested, and three variants (*bla*_OXA-1_, *bla*_OXA-2_ and *bla*_OXA-10_) were detected [82], while in WWTP active sludge in South Carolina, in the US, a higher variability among *bla*_OXA_ (seven variants) was noted [147]. Interesting results concerning the seasonal increase of *bla*_OXA_ concentration between the summer and winter seasons were reported in the study of four small-scale domestic WWTPs. Furthermore, *bla*_OXA_ in winter was prevalent among tested ARGs in raw sewage, as well as in effluent samples; additionally, the gene was detected in receiving river samples, in both the winter and summer seasons [84]. Results of a multi-national study, analyzing samples from Denmark, Spain, and the UK, indicated a country-specific presence for *bla*_OXA-10_ detected only in WWTPs from the UK [65].

The above data, showing the presence of *bla*_OXA_ genes and bacteria harboring them in WWTPs and related samples, confirms that WWTPs are a hotspot for antibiotic-resistant gene transmission into not only the aquatic compartments of the environment but also to the atmospheric air, creating an additional health risk for the workers of WWTPs.

**Table 1 ijerph-19-13829-t001:** ARGs encoding class A, B, C, and D β-lactamases detected in WWTPs-linked samples.

Location	Gene Variant(s) Detected in WWTP Samples ^1^				Sample Source(s) ^2^	Type of Tested Samples ^3^	Type of Methods ^4^	Ref.
	Class A	Class B	Class C	Class D				
Austria, Graz/Styria	CTX-M-15, -24 KPC-2 SHV-1, -26 TEM-1	nd	FOX	OXA-48	WWTP collecting DW and HW	CPE from SS	MM PCR seq-DNA	[95]
	CTX-M-1, -3, -14, -15, -38 PER-1 SHV-1, -2, -11, -12 TEM-1	nd	nd	nd		ESBL-producing *Enterobacterales* from SS		[81]
Austria	CTX-M TEM	nd	nd	nd	5 WWTPs	ESBL-producing *E. coli* from SS	MM PCR	[112]
Belgium, Brussels Capital Region	CTX-M KPC TEM	NDM-1	nd	OXA-48	2 WWTPs collecting HW	samples of IN, EF, RR and HW	qPCR	[89]
Czech Republic, Brno	CTX-M -1, 14b, -15 TEM-1	nd	nd	OXA-1	WWTP collecting HW	ESBL-producing *Enterobacterales* from EF	MM PCR seq-DNA molecular typing (MLST, PFGE)	[113]
Czech Republic, Moravian-Silesian Region	nd	NDM-1	nd	nd	2 WWTPs	samples from the nitrification and sedimentation tanks and bacteria isolated from them	MM qPCR WGS	[180]
France	CTX-M-1, -14, -15, -27 SHV-12	nd	nd	nd	WWTP collecting HW, rainwater	*E. coli* from IN, AS and EF	MM PCR seq-DNA molecular typing (MLST, PFGE)	[75]
Germany, Bielefeld-Heepen	CTX-M-4, -27, -32 GES-3 PER-2 SHV-34 TEM-1 TLA-2 VEB-1	IMP-2, -5, -9, -11, -13 VIM-4	AmpC CMY-5, -9, -10, -13	NPS-1, -2 OXA -1, -2, -5, -9, -10, -12, -18, -20, -22, -27, -29, -40, -45,-46, -48, -50, -54, -55, -58, -60, -61, -75	WWTP	strains from SS and EF, resistant to selected antibiotics	MM PCR seq-DNA	[136]
Germany, District of Kleve	CTX-M-1, -9 GES	VIM	ACT CMY-2 DHA FOX MIR	OXA-23, -48, -58	WWTP collecting DW, HW and IW	samples of IN, SS and EF, imipenem-, cefotaxime- or colistin-resistant strains	MM PCR, qPCR seq-DNA	[123]
Germany, North-Rhine Westphalia	IMI KPC	GIM VIM NDM	nd	OXA-48, -51	WWTPs collecting HW	ESBL-producing bacteria and CPE from HW, IN, EF, RR and rural wastewater	MM PCR molecular typing	[100]
Germany, South Region	nd	VIM-1	AmpC	nd	4 WWTPs collecting HW	samples of IN, EF and HW, receiving surface waters, groundwater and rain overflow	qPCR	[162]
Germany	CTX-M TEM	nd	CMY-2	nd	7 WWTPs with various inflow	IN and EF samples	qPCR seq-DNA	[135]
Ireland	CTX-M-1, -15 SHV-12 TEM-1l-like, -12, -116	nd	nd	nd	2 WWTPs	coliform strains from EF	MM PCR seq-DNA	[86]
Italy, The Oltrepò Pavese Plain	CTX-M-1, -14, -15, -28, -138 KPC-2 SHV-5 TEM-1	nd	nd	nd	4 WWTPs	cefotaxime-resistant *Enterobacterales* from WWTP, RR and groundwaters	MM PCR seq-DNA molecular typing (MLST, PFGE)	[78]
Poland, Kozieglowy	CTX-M KPC SHV TEM	NDM VIM	nd	OXA-1, -48	WWTP	samples of IN and EF, ESBL-producing and carbapenems-resistant coliforms	MM PCR qPCR seq-DNA	[69]
Poland, Olsztyn	CTX-M-1, -3, -9, -15 SHV-5 TEM-1, -47, -49	nd	nd	OXA-1	WWTP collecting HW	samples of IN, SS, EF, RR and the air near WWTP *Enterobacterales* from the samples	MM PCR seq-DNA	[68]
	SHV TEM	nd	nd	OXA		samples of IN, SS, EF, RR, nasal and throat employees’ swabs	metagenomics qPCR	[15]
	nd	IMP-1 VIM-2 NDM	nd	OXA-23, -24, -51, -58		*Acinetobacter* spp. from IN, SS, EF and RR	metagenomics qPCR	[163]
Poland, Warsaw	CTX-M-15, 27/98 GES-7 KPC PER-1/5, -3, -4 SHV-11, -12 TEM VEB	nd	ACC FOX-1, -2-like, -3, -4-like, -9, -10, -10-like, -13-like MOX-10/11, -4/8	OXA	WWTP collecting DW, MW and HW	*Aeromonas* spp. from IN, SS and EF	MM PCR seq-DNA	[97]
	CTX-M-1-like, -3-like, -15-like, -27-like GES KPC-2-like ORN PER-1/5 SHV-11-like, -12-like TEM-1-like, -12, -30, -47/68, -116	VIM-1-like, -2-like	ACT CMY-2-like, -4, -39, -40, -42/146/145, -65/75/89/113, -139, -157 FOX-15 MOX-13	OXA		ceftazidime- or meropenem- resistant Gram-negative bacteria from IN, SS and EF		[92]
Poland, Warmia and Mazury District	SHV TEM	nd	nd	OXA	4 domestic WWTPs	samples of IN, EF, and RR	qPCR	[84]
Poland, Warmia and Mazury District/Silesia District	Multiple variants i.e.,: AER BEL CARB Cfx GES TEM VEB	Multiple variants i.e., IMP VIM NDM	Multiple variants i.e.,: CMY FOX	Multiple variants i.e.,: LCR NPS OXA-23, -24, -48, -58	2 WWTPs collecting HW	samples of IN, SS and EF	metagenomics	[93]
Portugal, Coimbra	CTX-M -1, -9 TEM	nd	ACT MIR-1 FOX-1, -5 DHA-1, -2 CMY-2, -17, -12, -18, -21, -23 LAT-1, -3 BIL-1	OXA	WWTP collecting MW, HW and IW	*Enterobacterales* resistant to ampicillin from IN, EF, HW and RR	MM PCR	[88]
Northern Portugal	BEL-1 GES-5 TEM-1b	nd	CMY-101 DHA-1	OXA-1	WWTP collecting DW and HW	Gram-negative bacteria resistant to meropenem from IN, SS and EF	MM PCR qPCR seq-DNA molecular typing (rep-PCR, PFGE phylogrouping) WGS	[110]
Northern Portugal	CTX-M-1, -14, -15, -27, -32 SHV-1, -27 TEM-1	nd	nd	nd	WWTP	ESBL-producing and cefotaxime-resistant *Enterobacterales* from different stages of treatment	MM PCR seq-DNA	[118]
Romania, Cluj County	KPC-2	NDM-1, -6 VIM-2	nd	OXA-48	2 WWTPs, with and without hospital contribution	carbapenemase-producing *K. pneumoniae* from IN and EF	MM PCR molecular typing (MLST, phylogrouping)	[94]
Romania, Bucharest/Galati/Taˆrgovişte	CTX-M-15 KPC-2 SHV-1, -11, -12, -33, -100, -101, -106, -107, -145, -158, -187 TEM-1, -150	NDM-1	CMY-4 DHA-1	OXA-1, -9, -10, -48, -162	3 WWTPs HW	ESBL- and carbapenemase- producing *K. pneumoniae* from IN and EF	MM WGS	[85]
Slovakia, Kosice	CTX-M-1, -2	IMP	CMY-2	OXA-1	WWTP	ESBL-producing *E. coli* from IN and EF	MM PCR molecular typing	[115]
Spain, Catalonia	KPC	NDM	nd	nd	WWTP	samples of IN, EF, hospital EF, RR, sediment and biofilm	qPCR	[98]
Spain, Girona	TEM	nd	nd	nd	WWTP	samples of IN, EF, HW and RR	qPCR	[140]
Spain, Navarra	CTX-M-1, -14, -15, -55 SHV-12 TEM-1, 42, -145	nd	nd	nd	21 WWTPs	cefpodoxime-resistant *Enterobacterales* from EF	MM PCR seq-DNA	[76]
	nd	nd	ACC DHA EBC	nd	WWTPs	β-lactam-resistant bacteria from IN, EF and RR	MM PCR seq-DNA	[188]
Sweden, Stokholm/Uppsala/Lidingö	CTX-M GES KPC PER SHV SME TEM VEB	CAR IMP IND VIM	ACC ACT CFE CMY-1, -2 DHA FOX MIR MOX	OXA-1, -2, -10, 20, -23, -24, -48, -50, -51, -58, -60, -63	3 WWTPs collecting MW, HW, IW and storm water	samples of IN, SS and EF	metagenomics	[96]
Sweden, Örebro	CTX-M-1, -9 GES PER-1 SFO-1 SHV VEB	IMP-5, -12	ACC-1, -3 ACT-1, -5/7 CFE-1 CMY-10 DHA FOX LAT MIR MOX	OXA -2, -10, -50, -51, -58	WWTP collecting DW, HW and IW	Gram-negative bacteria from IN, EF, HW, RR and lake water	MM qPCR	[109]
Sweden, Stockholm	CTX-M-1, -9	nd	CMY-2	nd	WWTP	*E. coli* from IN, EF and HW	MM PCR	[116]
Switzerland, Basel	KPC-2	NDM-1, -5 VIM-1	nd	OXA-48, -181	WWTP collecting MW and HW	CPE and Gram- negative bacteria from IN, EF, HW and RR	MM PCR seq-DNA molecular typing (MLST, phylogrouping)	[99]
The UK	CTX-M-15 LEN-25-like OXY-6 SHV-12 TEM-1	IMP-1 NDM-1-like, -5	nd	OXA-1, -17, -48, -181	20 WWTPs	carbapenem-resistant Gram-negative strains isolated from treated and untreated samples	MM WGS	[128]
	CTX-M-1, -14, -15, -27	nd	nd	nd		ESBL-producing *E. coli* from treated and untreated samples	MM metagenomic	[120]
Canada, Arnpior/Ottawa/Toronto	CARB CTX-M GES KPC OXY PER SHV TEM	cphA IMP VIM PAM	ACT CepH FOX, MOX	OXA-2, -10	3 WWTPs	carbapenem-resistant strains from IN	MM PCR, WGS	[104]
Canada, Alberta/Calgary	CTX-M-15	nd	AmpC	OXA-1	13 WWTPs	multidrug-resistant *E. coli* from IN and EF	MM PCR WGS molecular typing	[133]
Canada, Baffin Island (Pond Inlet/Clyde river/Iqaluit)	CTX-M TEM	nd	nd	nd	3 WWTPs	IN and EF samples	qPCR	[139]
Guadeloupe/North America	CTX-M-1, -8, -14, -15, -27 TEM-1-like, -3 VEB-1	nd	CMY-2, -8	OXA-1-like	2 WWTPs	*Enterobacterales* from IN, EF, RR and sea waters, with a focus on ESBL- and AmpC-producers	MM PCR seq-DNA phylogrouping	[80]
The US, Colorado	CTX-M TEM	nd	nd	OXA-1	WWTP	ESBL- and KPC-producing *E. coli* from IN and EF	MM PCR seq-DNA WGS	[119]
The US, South Carolina	BES-1 CTX-M-1 GES KPC SHV TLA-1 VEB	ccrA IMP-5, -12	ACT-1 CMY-10 FOX LAT MIR MOX	OXA-2, -10, -23, -24, -51, -58, -60	WWTP	samples of SS and bioaerosol collected downwind from sludge aeration tanks and upwind from WWTP	MM qPCR seq-DNA	[147]
The US, Washington	CTX-M KPC TEM	NDM-1	CMY-2	OXA-48	2 WWTPs	samples of IN, SS, EF, RR and irrigation water	PCR	[90]
The US, Wisconsin	CTX-M-1 and -9 group TEM	nd	nd	OXA	WWTP	cefotaxime-resistant *E. coli* from IN, EF and HW	MM PCR molecular typing WGS	[125]
The US (New Jersey, Maryland, Ohio, Texas, Colorado, California)	CTX-M GES KPC TEM	VIM NDM	nd	OXA	7 WWTPs with various inflow	*E. coli* from EF	MM PCR seq-DNA molecular typing (phylogrouping, sequence typing)	[103]
The US	CARB-2 CTX-M-15 GES-5 KPC-2, -3 OKP-B-2, -7 ORN-1b OXY-1, -5 PLA-2 SHV-11, -12 TEM-1, -1a, -1b	VIM-1 NDM-1, -5, -7	AmpC ACT-1 CMY-66, -79 FOX-5 MIR-3, -6, -9, -15	OXA-1, -2, -9, -105	50 WWTPs	carbapenemase-producing bacteria from EF and surface water of WWTP discharge	MM WGS	[102]
Colombia, Antioquia	CTX-M-1, -2, -8/25, -9 SHV TEM	nd	LAT/BIL/CMY group ACT/MIR group DHA	nd	WWTP collecting DW, HW and IW	β-lactam-resistant Gram negative bacilli from IN, SS and EF, with focus on *E. coli*	MM PCR seq-DNA molecular typing (PFGE, MLST)	[87]
Brazil, Curitiba	CTX-M-1, -2, -8, -9, -15 SHV-12 GES-5	nd	nd	nd	WWTP	cefotaxime-resistant Gram-negative bacteria from IN, SS, EF, hospital, sanitary effluent and RR	MM PCR seq-DNA	[79]
Brazil, São Paulo	CTX-M-8, -15 SHV-28	nd	nd	nd	5 WWTPs	ampicillin-resistant *Enterobacterales* from IN	MM PCR seq-DNA molecular typing (phylogrouping *E. coli*, MLST)	[117]
India, Jasola Vihar, New Delhi	CTX-M-15, -152, -205 SHV TEM-1	nd	nd	nd	WWTP	ESBL-producing bacteria from EF, lentic water bodies and slaughterhouse	MM PCR seq-DNA	[71]
India, New Dehli	nd	NDM-1	nd	nd	12 WWTPs	coliforms bacteria from EF	MM PCR seq-DNA	[183]
India, Jaipur	Multiple variants i.e.,: Cfx-A2, -A3 GES-15 VEB-1	nd	nd	NPS-1 OXA-209	4 WWTPs collecting HW	samples of IN, SS and EF	metagenomics	[111]
India, State of Bihar, Goa, Karnataka, Tamil Nadu and Telangana	CTX-M-15, -55 SHV-12 TEM-1, -1b	NDM-1, -5, -7	CMY-2, -6, -42	OXA-1, -9, -10	5 WWTPs	ESBL- and carbapenem- producing *E. coli* from WWTPs and rivers	MM PCR seq-DNA, molecular typing WGS	[77]
Singapore	cfxA6 TEM VEB-1a	nd	AmpC	OXA-198, -333, -347	WWTP	samples of IN, EF, HW and surface waters	metagenomics	[108]
Singapore	AER-1 CARB-3, -(5-9), -12 Cfx-A2, -A3 CTX-M-1, -15, -19, -34, -147 KPC-1, -10, -11, -13, -16 LEN-19, -21 OKP-A, -B PER-1, -3, -4, -7 PSE-1, -4 ROB-1 SHV-4, -12, -39, -51, -53, -167 VEB-(2–8) multiple variants of GES and TEM groups	GOB-1 IMP-31 LRA	ACT-2, -3, -16, -19, -20 DHA-6, -5, -6, -7 FOX-2, -4, -5, -7, -8, -9 MIR-1, -2, -6, -8 MOX-(1–7) PDC-2, -5 multiple variants of CMY group	LCR OXA-278	WWTP	samples of IN, SS and EF	metagenomics	[107]
China, Guangdong Province	nd	nd	AmpC	nd	2 WWTPs	*E. coli* from WWTPs	MM PCR	[190]
China, Harbin	CTX-M	nd	nd	nd	4 WWTPs	samples of IN, SS and EF	PCR qPCR	[137]
China, Tianjin	KPC-2 GES-1	nd	nd	nd	WWTP collecting DW and IW	EF samples	qPCR	[106]
	nd	NDM-1	nd	nd		EF and RR samples	qPCR	[178]
China, Wuxi	CTX-M SHV TEM	nd	nd	nd	3 WWTPs collecting DW and IW	IN and EF samples, cultivable heterotrophic bacteria and total coliforms	MM qPCR seq-DNA	[129]
China	CTX-M TEM	VIM	nd	nd	3 WWTPs	multiple antibiotic-resistant *Escherichia* spp. from WWTPs, HW and livestock manure	MM PCR seq-DNA	[70]
China	nd	NDM-1	nd	nd	2 WWTPs collecting DW and IW	samples of IN, SS and EF	MM PCR seq-DNA	[177]
Japan, Tokyo	CTX-M-1 group, -2 group and -9 group SHV group TEM group	nd	nd	nd	WWTP	fecal coliforms from different stages of treatment process	MM PCR seq-DNA	[124]
Japan	CTX-M-1, -2, -3, -8, -14, -15, -27, -55, -64, -65, -123, -174	nd	nd	nd	4 WWTPs	cefotaxime-resistant *E. coli* from IN	MM PCR seq-DNA molecular typing (MLST, phylogrouping) WGS	[122]
United Arab Emirates, Dubai	SHV TEM	nd	nd	nd	WWTP	ESBL-producing *Enterobacterales* from SS	MM PCR	[83]
Saudi Arabia, Jeddach	nd	NDM-1	nd	nd	WWTP	ESBL- and carbapenemase- producing bacteria from IN	MM qPCR WGS	[179]
South Africa, Durban	CTX-M TEM	nd	nd	nd	WWTP collecting DW, HW and IW	ESBL-producing *E. coli* from IN, SS, EF and RR	MM PCR	[130]
	CTX-M KPC-2 TEM	NDM-1	nd	OXA-1		coliforms bacteria from IN and EF focused on *E. coli*	MM PCR	[91]
South Africa, Mgungundlovu District	CTX-M-3, -15, -28 SHV-28 TEM-1, -116, -181, -213, -215	nd	nd	nd	4 WWTPs collecting DW, HW and IW	ESBL-producing *E. coli* from IN and EF	MM PCR seq-DNA	[73]
South Africa, Eastern Cape Province, Amathole District	PSE-1 TEM	nd	nd	nd	2 WWTPs	*Aeromonas* spp. from WWTPs	MM PCR	[126]
South Africa, Eastern Cape Province, Amathole and Chris Hani District	CTX-M-1, -2, -9 GES KPC PER SHV TEM	nd	ACC CIT DHA EBC MOX	OXA-1-like, -48-like	2 WWTPs collecting DW, IW, run-off waters and residential sewage	*Enterobacterales* from EF	MM PCR	[72]
South Africa, Eastern Cape Province, Amathole, Chris Hani and Sarah Baartman District	KPC	NDM-1	nd	nd	4 WWTPs	*Enterobacterales* from EF, HW and surface waters, with focus on *Klebsiella* spp.	MM PCR	[105]
Algeria, Boumerdes	CTX-M-3, -15 TEM-1	nd	nd	OXA-1	WWTP collecting DW, HW and IW	cefotaxime-resistant strains from IN and EF	MM PCR seq-DNA, molecular typing (MLST, phylogrouping)	[74]
Tunisia	CTX-M-1, 14a, -15 TEM-1a, -1b	nd	CMY-2	OXA-1	8 WWTPs	cefotaxime-resistant *Enterobacterales* from IN, EF, MW, effluents of MW and IW, RR and surface waters not connected to WWTP	MM PCR seq-DNA molecular typing of *E.coli* (MLST, phylogrouping, PFGE, virulence genotyping)	[127]
Tunisia	CTX-M-1, -3, -14, -15, -27	nd	nd	OXA-204	2 WWTPs	ESBL-producing *Enterobacterales* from WWTP and various animal samples	MM PCR seq-DNA, molecular typing (MLST, phylogrouping, PFGE)	[121]
Tunisia, Monastir Governorate	CTX-M TEM	nd	nd	nd	5 WWTPs collecting DW, HW and IW	IN and EF samples	qPCR	[142]
Australia, Queensland	CTX-M TEM	nd	nd	nd	2 WWTPs	ESBL-producing *E. coli* from IN and HW	MM PCR molecular typing	[67]
Multinational study: Denmark, Spain, the UK	cfxA BEL CARB CTX-M-1, -3, -15 GES KPC LEN OXY-1, -2 SFO SHV-11 SPM TEM TLA VEB	IMP VIM NDM	AmpC ACC CMY DHA FOX MIR	OXA-10	3 WWTPs collecting DW and HW	samples of IN, SS, EF and RR	qPCR seq-DNA	[65]
Multinational study: Finland, Estonia	CTX-M-32 SHV-34	nd	nd	OXA-58	3 WWTPs	IN and EF samples	qPCR	[141]
Multinational study: France, Italy, Norway, Portugal, Germany, Netherlands, Cyprus, Turkey, Austria and the UK	CTX-M-15, -32 KPC-3 TEM	nd	nd	OXA-48, -58	16 WWTPs	samples of EF and corresponding receiving water bodies	qPCR	[101]
Multinational study: China, Singapore, the US, Canada	TEM-1	IMP	AmpC	OXA-1, -2, -10	15 WWTPs	SS samples	PCR qPCR	[82]

^1^ nd—no data. ^2^ WWTP—wastewater treatment plant, DW—domestic wastewater, MW—municipal wastewater, HW—hospital wastewater, IW— industrial wastewater. ^3^ CPE—carbapenem-resistant *Enterobacterales*, IN—influent, EF—effluent, SS—sewage sludge, RR—receiving river waters. ^4^ MM—microbiological methods, PCR—specific PCR, qPCR—quantitative PCR, seq-DNA—sequencing DNA, WGS—whole genome sequencing, MLST—multilocus sequence typing, PFGE—pulsed-field gel electrophoresis.

## 5. Conclusions

AMR is a serious and urgent problem, and it is clear that the environment plays a key role in the process of transmission and propagation of ARGs and ARB with life-threatening clinical consequences. The multitude of publications confirms that β-lactamases genes encoding especially ESBLs (TEM, SHV, CTX-M) and KPC, NDM, and OXA carbapenemases, which pose one of the greatest health risks, are widely found in WWTPs and disseminated to further portions of the environment. Molecular analysis shows repeatedly high genetic relatedness between environmental and clinical isolates, e.g., ST131 *E. coli*. Generally, different kinds of sewage treatment processes do not eliminate these ARGs completely. Furthermore, some data indicate an increased level of β-lactam ARGs in effluent or even the presence of the genes and bacteria harboring them in samples after additional disinfection treatments.

Due to β-lactam ARGs’ potential to transfer via mobile genetic elements through horizontal gene transfer, their abundance in water samples discharged from WWTPs into natural aquatic sources used by humans or animals suggests a potential risk of transmission resistance determinants into pathogenic and non-pathogenic bacteria and acquiring multidrug resistance as well as the participation of WWTPs in AMR transmission route and distribution into surrounding ecosystems and clinical settings. The growing problem of AMR and the spread of clinically relevant ARGs related to, i.e., β-lactams in the environment, indicate the need to improve and evaluate the procedures of wastewater treatment and disinfection; thus, ARB, ARGs, and factors influencing their selection and co-selection during the treatment process would be completely removed.

The development and improvement of techniques used in testing wastewater for antibiotic resistance has been very significant in recent years. There are more and more publications indicating the use of modern metagenomic assays, which enables broadening the knowledge of the complexity and structural and functional biodiversity of microbial communities—i.e., analysis of resistance genes; taxonomic assignment; functional genes characterization; the identification of the HGT mechanism and mobile elements involved in the gene transmission; and exploring relationships between pathogenic and non-pathogenic species and susceptible and resistant bacteria. Therefore metagenomic analysis seems to be a very useful tool to understand the process of AMR transmission. However, the clinical surveillance of resistant strains responsible for life-threatening infections and nosocomial outbreaks caused by β-lactam-resistant strains also involve molecular techniques, but still the gold standard are culture-based methods detecting the expression of genes and the resistance mechanism. Therefore, according to the One Health’s concept, collaborative approaches concerning AMR in the environment and clinical setting are indispensable and should combine new technology with standard microbiological methods. As WWTPs are the crucial points on the routes of ARB and ARGs’ spread, they should be deeply explored, which would help to understand the process and make it possible to introduce procedures to stop, or at least slow down, the spreading of antibiotic resistance.

## Figures and Tables

**Figure 1 ijerph-19-13829-f001:**
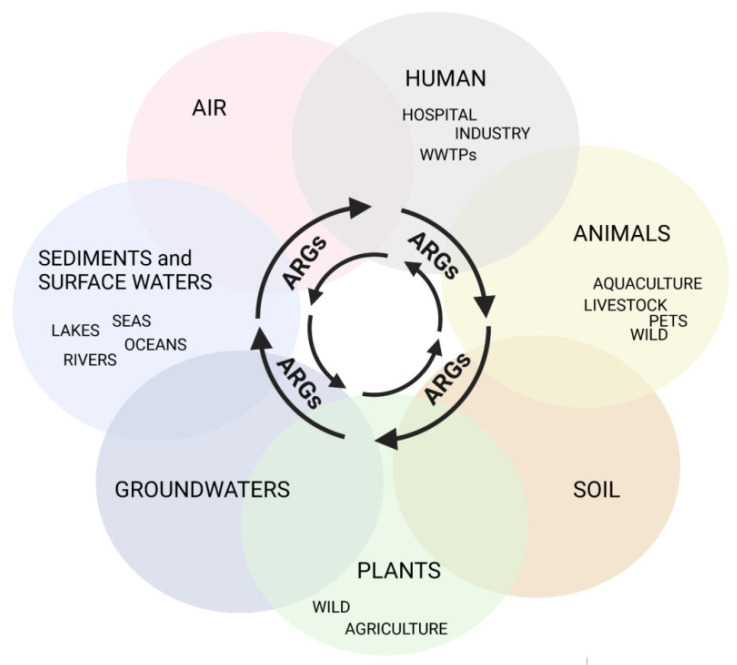
Routes of ARG transmission in the total environment, created with BioRender (available online: https://biorender.com, accessed on 23 May 2022).

**Figure 2 ijerph-19-13829-f002:**
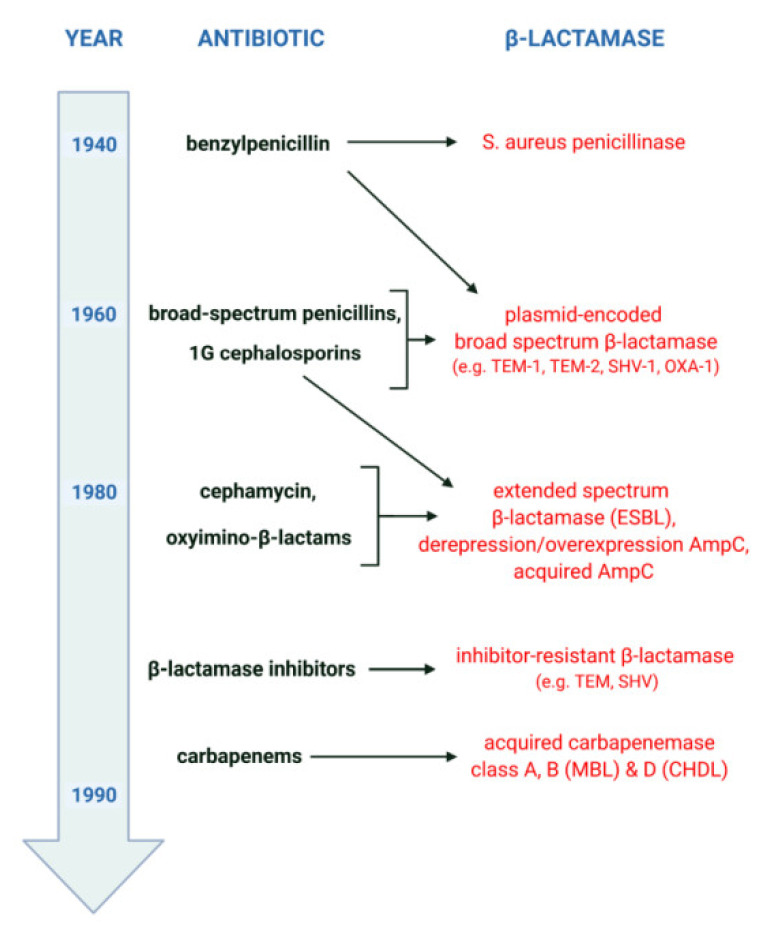
Timeline of the evolution of β-lactamases, created with BioRender (available online: https://biorender.com, accessed on 23 May 2022).

**Figure 3 ijerph-19-13829-f003:**
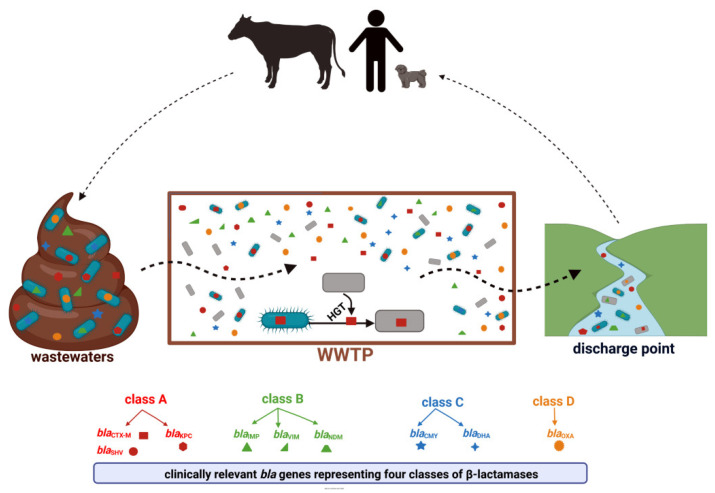
WWTP as a hotspot for the transmission of clinically relevant β-lactam resistance genes, created with BioRender available online: https://biorender.com, accessed on 23 May 2022). Descriptions of the enzymes included in the Figure 3 can be found in Section 4.1, Section 4.2, Section 4.3 and Section 4.4.

## Data Availability

Not applicable.

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
