# Peer review of "Clinically Relevant β-Lactam Resistance Genes in Wastewater Treatment Plants"

_ijerph, 2022, doi:10.3390/ijerph192113829_

Round 1
Reviewer 1 Report
The present manuscript is a detailed review of the literature on the analysis of beta-lactamase genes in wastewater treatment plants. The topic is certainly relevant and interesting to a wide range of specialists in the field of studying the spread of antibiotic-resistant bacteria and environmental protection. The review shows in detail that WWTPs represent the hotspots for ARGs transmission into the aquatic environment and the atmospheric air. In addition to the analysis of the prevalence of beta-lactamase genes of different types, characterized by different specificity in relation to beta-lactam groups, the review contains the data on the removal of selected resistance genes during the treatment process. Threatening are the facts that the genes of some enzymes are found after processing.
The review is well written, detailed enough to understand the state of the problem, and includes an analysis of a large amount of experimental data.
In my opinion, the text is almost ready for publication, I have only minor comments:
In my opinion, it would be better to use the plural for beta-lactamases in Fig. 2 because of their diversity.
Line 116 Is association of beta-lactamase inhibitors with beta-lactam antibiotics justified? Inhibitors of the 1st generation, having a beta-lactan ring in the structure, have different antibacterial activity. New generations of inhibitors do not have a beta-lactam ring and differ in structure from beta-lactams.
Author Response
Dear Reviewer,
Thank you very much for your comments.
In our opinion, the description in the figure 2 does not require the use of plural, for example concerning benzylopenicillin only one type of S. aureus beta-lactamase is produced.
Concerning line 116 -
Obviously it is our mistake to include inhibitors of beta-lactamases into group of beta-lactamases. It has been corrected in the text.
Your sincerely
Izabela Waśko

Reviewer 2 Report
The manuscript reviews the presence of clinically relevant b-lactam resistance genes in wastewater treatment plants. However, the authors also describe their presence in river and other types of waters, so change the name to environmental waters. the manuscript is mostly well-organized but needs moderate english revision. Comments:
- page 4, lines 148-149: how can I take this affirmation from Fig.2?
- page 5, line 192: write PCR in full first. this is a common problem with the manuscript in which the authors use abbreviations (see for instance page 7, line 278 - E. coli) without writing first in full;
- page 8, line 371: blaGES write in italic;
- page 8, lines 348 - 349: add references;
- page 10, lines 417-418: why have the authors chosen to present the presence of the genes like that? Enterobacterales is an order, whereas Pseudomonas and Acinetobacter are genus. Especially, because the last two belong to the same order.
- page 13, lines 581 - 582: do the authors mean that bacteria producing OXA enzymes survive to treatment and that the gene is also found following treatment? If so, please rectify the sentence as it looks like genes are alive and therefore have the capability of surviving;
- page 13, line 592: "comparing raw and treated samples", are these from WWTP? if so, please add such information. Water entering drinking water treatment plants is also raw, and when they leave the DWTP, it is treated;
- page 31, lines658 to 660: how do the authors think that advancements in metagenomics will help in understanding the process of AMR transmission?
Author Response
Dear Reviewer
We are very appreciate for your valuable comments and for catching inaccuracies. We have corrected them as you suggested. Concerning changing the title - our opinion is quite different and explanation is included in the attached file.
Your sincerely
Izabela Waśko
